# Identification of functional tetramolecular RNA G-quadruplexes derived from transfer RNAs

Shawn M. Lyons[1,2], Dorota Gudanis[3], Steven M. Coyne[1,2], Zofia Gdaniec[3] & Pavel Ivanov [1,2,4]

RNA G-quadruplex (RG4) structures are involved in multiple biological processes. Recent genome-wide analyses of human mRNA transcriptome identified thousands of putative intramolecular RG4s that readily assemble in vitro but shown to be unfolded in vivo. Previously, we have shown that mature cytoplasmic tRNAs are cleaved during stress response to produce tRNA fragments that function to repress translation in vivo. Here we report that these bioactive tRNA fragments assemble into intermolecular RG4s. We provide evidence for the formation of uniquely stable tetramolecular RG4 structures consisting of five tetrad layers formed by 5′-terminal oligoguanine motifs of an individual tRNA fragment. RG4 is required for functions of tRNA fragments in the regulation of mRNA translation, a critical component of cellular stress response. RG4 disruption abrogates tRNA fragments ability to trigger the formation of Stress Granules in vivo. Collectively, our data rationalize the existence of naturally occurring RG4-assembling tRNA fragments and emphasize their regulatory roles.

[1] Division of Rheumatology, Immunology and Allergy, Brigham and Women's Hospital, Boston, MA 02115, USA. [2] Department of Medicine, Harvard Medical School, Boston, MA 02115, USA. [3] Institute of Bioorganic Chemistry, Polish Academy of Sciences, Noskowskiego 12/14, 61-704 Poznan, Poland. [4] The Broad Institute of Harvard and M.I.T, Cambridge, MA 02142, USA. Shawn M. Lyons and Dorota Gudanis contributed equally to this work. Correspondence and requests for materials should be addressed to Z.G. (email: zgdan@ibch.poznan.pl) or to P.I. (email: pivanov@rics.bwh.harvard.edu)

G-quadruplexes (G4s) are guanine-rich nucleic acid structures composed of stacked planar G-quartet motifs, in which four guanine molecules interact with each other through Hoogsteen hydrogen bonding (Fig. 1a)[1]. Monovalent cations can stabilize G4 structures by coordinating the O6 atom in the channel of G-quartet. Whereas certain ions such as physiologically relevant $Na^+$ and $K^+$ cations with adequate ionic radii promote G4 assembly, those ions with smaller ionic radii, such as $Li^+$, do not support G4 formation[2,3]. G4s are very stable under various conditions typically sufficient to unfold other nucleic acid structures, such as increased temperature or chemical denaturation (e.g., by urea and formaldehyde).

Biologically relevant G4s readily form in vitro and are unusually stable once formed. DNA G4s exist in vivo and are implicated in a wide range of diverse biological processes including transcription regulation, DNA recombination, genome stability and telomere maintenance[4–6]. For instance, telomeric DNA consists of the sequence "TTAGGG", tandemly repeated up to 100,000 times in mammalian genomes, assembling G4s that can be visualized at telomeres by G4-specific antibodies in live cells[7,8].

While much effort has been brought into the studying of DNA G4s[9,10], RNA is equally capable of assembling into RNA G4s (RG4s)[11,12]. As RNA exists in a single-stranded state in cells, it is conceivable that it is more likely to form G4s as this structure does not have to compete with a double-stranded helix. Unsurprisingly, the *TERRA*, non-coding RNA (ncRNA) which arises from transcription of telomeric DNA, assembles into RG4s[13]. RG4s have also been identified within mRNAs and they are enriched within the untranslated regions (UTRs), regions playing various regulatory roles in mRNA metabolism such as mRNA translation[7]. Although some RG4s have been shown to stimulate translation[14], most reports indicate that RG4s in mRNAs may function to inhibit translation acting as a roadblock for the scanning pre-initiation/ribosomal complexes or by attracting known translation inhibition factors[15–17]. Transcriptome-wide analysis indicates that thousands of mRNAs assemble RG4s in vitro[18,19]; however, the same RG4s are disassembled in living cells[18]. The biological significance of RG4s unfolding in vivo is largely unknown but it is suggested that their resolving by specific proteins, such as CNBP, stimulates/supports mRNA translation[20].

While roles of RG4s in mRNA metabolism are under active investigation, the role of RG4s in small ncRNAs is just emerging. We have shown previously that mature cytoplasmic tRNAs are a source of a small RNAs which we termed tRNA-derived stress-induced RNAs (tiRNAs)[21]. tiRNAs are produced in response to various cellular stresses (e.g., oxidative stress, UV irradiation, and nutrient starvation) through RNA cleavage mediated by angiogenin (ANG), a stress-responsive member of the ribonuclease

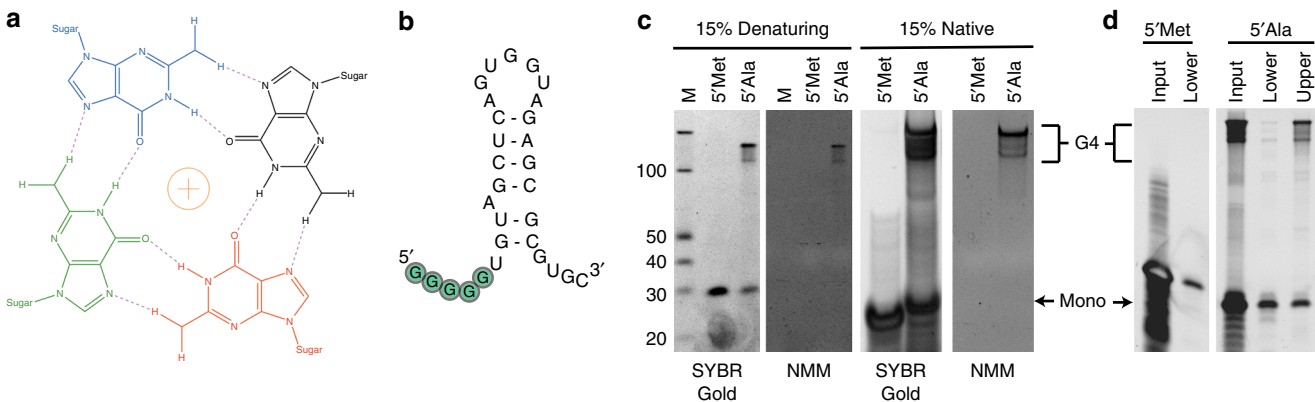

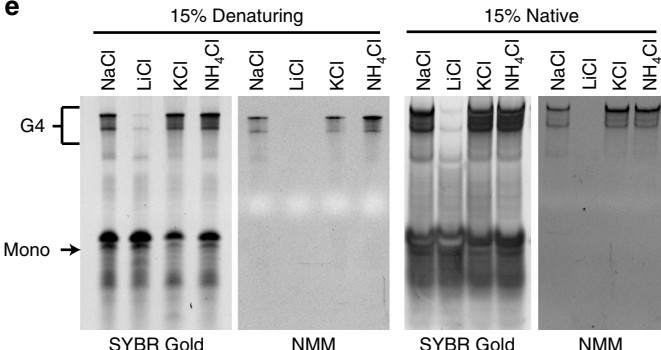

**Fig. 1** 5′TOG containing tiRNAs exist in two distinct and interconvertible forms. **a** G-quartet formation by guanines. **b** Predicted secondary structure of 5′tiRNA^Ala with a predominant hairpin structure. 5′TOG motif is indicted in green. **c** Indicated RNAs were ran on a 15% acrylamide denaturing gel (left panels) or native gel (right panels). Gels were post-stained with either SYBR gold to detect total RNA or NMM to detect RG4 structures. 5′tiRNA^Ala ran at the expected 31 nts, but also at ~120 nts. Only the more slowly running band stained with NMM, indicating it contained RG4s. This more slowly migrating band persisted under denaturing conditions. **d** Indicated 5′tiRNAs were ran on a 15% denaturing acrylamide gel and either upper or lower species were gel purified and reanalysed on a denaturing acrylamide gel and post-stained with SYBR gold. A portion of the lower band re-equilibrated to the upper band and vice-a-versa. **e** 5′tiRNA^Ala was equilibrated overnight in 100 mM of indicated salts. RNAs were analyzed on denaturing (left panels) or native gels (right panels) and post-stained with SYBR gold or NMM. Results demonstrate that NMM staining, more slowly migrating band disassembles in the presence of $Li^+$ ions

(RNase) A superfamily[22,23]. ANG cleaves mature tRNAs within the anticodon loop to produce 5′ and 3′ tiRNAs[24]. A subset of 5′ tiRNAs derived from tRNA[Ala] and tRNA[Cys], (5′ tiRNA[Ala] and 5′ tiRNA[Cys], respectively) inhibits protein synthesis by interfering with cap-dependent mRNA translation initiation. Mechanistically, these 5′ tiRNAs target cap-binding eIF4F complexes to displace them from the cap (m7GTP) structures of mRNAs[25]. Consequently, inhibition of translation initiation leads to the formation of stress granules (SGs), pro-survival cytoplasmic foci containing stalled pre-initiation ribosomal complexes[26–28]. The ability of these tiRNAs to repress translation and promote SG formation is dependent upon a stretch of guanine residues at their 5′ ends, which has been termed a 5′ terminal oligoguanine (5′ TOG) motif (Fig. 1b)[25,29,30]. Our initial studies suggested that the 5′TOG motif may assemble into a "G-quadruplex-like structure"[30]. Here we demonstrate the functional requirement of the 5′ TOG motif. The stretch of 5 guanosine residues within the 5′TOG interacts the 5′ TOG motifs of other tiRNAs, forming a tetramolecular RG4. We demonstrate that this structure is required for 5′ tiRNA activity, rather than simply RNA-protein interactions driven by sequence determinants. Further, we demonstrate that production of functional, tetrameric, RG4-assembling tRNA-derived fragments is an evolutionary conserved process. Our data highlight importance of intermolecular RG4 in post-transcriptional regulation of gene expression.

## Results

**5′tiRNA[Ala] assembles two alternative structural forms**. Previously, we identified several molecular species of 5′ tiRNA[Ala] that migrate aberrantly in native and denaturative gels and hypothesized that some of those species may assemble G4-like structures[30]. Indeed, 5′tiRNA[Ala] exhibits a distinct and aberrant migration pattern in comparison to 5′tiRNA[Met], a non-5′TOG-containing 5′tiRNA of the same size (Fig. 1c). On a denaturing gel stained with SYBR Gold, a dye detecting nucleic acids, 5′tiRNA[Met] demonstrates mobility of the expected 31 nts. While a fraction of 5′tiRNA[Ala] also ran at 31 nts, a second band running at the size of 120 nts was also observed. The same phenomenon was also confirmed in native gels (Fig. 1c, right panels). We reasoned that the guanine residues in the 5′TOG-motifs of individual 5′tiRNA[Ala] molecules might interact to form an intermolecular RG4 complex assembled by four individual 5′tiRNA[Ala] strands explaining the observed migration of approximately ×4 the size of the expected monomer and the stability in the urea denaturant. To test this hypothesis, we stained native and denaturing gels with N-methyl mesoporphyrin IX (NMM), a molecule that has strong affinities for RG4 structures and fluoresces when excited by UV light[31]. The upper more slowly migrating band in 5′tiRNA[Ala] stains with NMM, while 5′tiRNA[Met] or the lower 5′tiRNA[Ala] band does not (Fig. 1c) supporting the hypothesis that the upper band is an RG4 structure and the lower band is a monomeric form of 5′tiRNA[Ala]. To exclude the possibility that the NMM staining band was a contaminant generated during chemical synthesis of 5′tiRNA[Ala], we ran both 5′tiRNA[Met] and 5′tiRNA[Ala] on denaturing gels and gel purified both bands separately (Fig. 1d). The resulting gel purified RNAs were re-analyzed on a denaturing acrylamide gel and visualized by SYBR Gold staining. When gel purified, a portion of the lower monomeric band converts into the upper more slowly running species and the converse was also true. These data demonstrate that, rather than being two distinct RNA species, these bands are two possible structural forms of 5′tiRNA[Ala].

If the upper more slowly migrating band is a tetramer tethered by an RG4, we reasoned that this structure can be manipulated through ionic equilibration with monovalent cations modulating

RG4 formation. Those ions with radii large enough to be efficiently coordinated ($K^+$, $Na^+$, and $NH_4^+$) support RG4 formation. Conversely, if the ionic radii are too small to be coordinated (e.g., $Li^+$), RG4 structures will be depleted. Therefore, we equilibrated 5′tiRNA[Ala] in various monovalent salts and analyzed the mobility on native or denaturing gels stained with SYBR gold to monitor total nucleic acid or NMM to monitor RG4 structures (Fig. 1e). In agreement with the upper band being a RG4 5′tiRNA[Ala] tetramer, equilibration in $Li^+$ led to collapse of the upper band to the monomeric form and loss of staining with NMM. In total, the apparent size, the ability to interconvert, the staining with NMM and the dependence upon specific ions strongly support the idea that 5′tiRNA[Ala] can exist in monomeric and tetrameric forms, where the latter may adopt RG4 structure.

**Equilibrium between RG4 and hairpin structures in 5′tiR-NA[Ala]**. NMM staining assay is instrumental but not definitive to state that an RNA molecule adopts RG4 structure. To prove that 5′tiRNA[Ala] can fold into RG4 structure and to obtain information on its topology in different solution conditions, we further employed biophysical approaches. NMR spectroscopy is a powerful tool to study RNA secondary structure since the imino proton "fingerprint" region of NMR spectra between 10 and 15 p. p.m. can provide information about the types of base pair formed. Imino proton peaks from 12 to 15 p.p.m. are characteristic of Watson–Crick base pairs, whereas those from 10 to 12 p.p.m. are associated with unpaired or unstacked residues and are generally broad and their observation requires working at low temperature[32]. Imino resonances from Hoogsteen base pairs, characteristic of G-quartet formation, are also found in 10–12 p.p.m. region. Figure 2a shows the imino region of the [1]H NMR spectra of 5′tiRNA[Ala] obtained in the presence of monovalent ions $K^+$, $Na^+$, or $Li^+$. In the spectrum recorded in a buffer containing $Li^+$ ions (Fig. 2a) three sharp signals are observed at 14.0, 13.4, and 11.8 p.p.m., each corresponding to one proton, broader signal at 13.0 p.p.m. arising from two protons and two smaller signals at 11.7 and 11.1 p.p.m. Overall, the spectra obtained in the presence of $K^+$, $Na^+$, or $Li^+$ ions (Fig. 2a) are very similar in the region corresponding to Watson–Crick base pairs. Highfield resonances at 11.7 and 11.1 p.p.m. also appear in spectra recorded in the presence of $K^+$ or $Na^+$, although with different intensities.

The most thermodynamically stable secondary structure predicted for 5′tiRNA[Ala] by RNA structure software package[33] is a hairpin depicted in Fig. 1b. Generally, this structure agrees with NMR data where the presence of four G-C and one A-U Watson–Crick base pairs was identified using 2D NOESY spectrum. Additionally, the appearance of a strong cross peak between two highfield resonances at 11.8 and 11.1 p.p.m. in 2D NOESY spectrum (Supplementary Fig. 1a) suggested the formation of a G-U base pair. Different intensities of these resonances in the three [1]H NMR spectra most likely reflect different G-U base pair dynamics under the conditions utilized.

Next, we have focused our attention on the region between 10.5 and 12 p.p.m. corresponding to the Hoogsteen base pairs region. Very broad signals appeared in the spectra recorded in buffers containing $K^+$, $Na^+$ but not $Li^+$ ions suggesting that they may originate from the RG4 form of 5′tiRNA[Ala]. We have recorded [1]H NMR spectra as a function of the temperature to examine whether the species of 5′tiRNA[Ala] present in solution differ in their thermal stability as it is known that RG4s are distinguished among other nucleic acid structures by significant thermal stability. The NMR melting profiles of 5′tiRNA[Ala] in the temperature range from 15 to 55 °C for samples recorded in

buffers containing K⁺, Na⁺, or Li⁺ ions are shown in Fig. 2b, respectively. Broad signals observed in ¹H NMR spectra recorded in K⁺ or Na⁺ buffers progressively sharpen in the Hoogsteen base pairs region as the temperature increases to 55 °C. At this temperature only signals around 11 p.p.m., which is characteristic of RG4 structure, are still clearly visible, while the other are completely undetectable. However, when ¹H NMR spectrum of 5′tiRNA$^{Ala}$ was recorded at 55 °C in the presence of lithium ions (Fig. 2b) no imino signals could be observed. The results of these experiments support the existence of RG4 structures in K⁺ and Na⁺ but not in Li⁺ solution.

One of the characteristic features of RG4s is significant solvent protection of the RG4 core that correlates with its very high thermodynamic and kinetic stability[34,35]. In ¹H NMR spectra, it manifests itself as the slow disappearance of imino protons after solvent exchange from H₂O to D₂O. In RG4s, a complete replacement of the hydrogen bonded atoms with deuterium generally occurs after many hours, sometimes even after a few weeks or months. This is quite unlike DNA or RNA duplexes or triple helices, where the imino protons typically exchange in seconds or minutes. Figure 2c shows the comparison of the spectra of 5′tiRNA$^{Ala}$ recorded in Na⁺ (left panel) and K⁺ (right panel) containing solutions at 45 °C in 90% H₂O/10% D₂O and after the exchange to D₂O. In the spectrum recorded in D₂O in the presence of potassium ions, signals attributed to the RG4, are still observed. However, there is only one resonance left in the NMR spectrum of the sample of 5′tiRNA$^{Ala}$ in Na⁺ buffer after the exchange to D₂O. This demonstrates the lower RG4 stability in the presence of Na⁺ than K⁺ ions, in agreement with stronger preference of RG4s for K⁺ over Na⁺ ions.

G-quadruplexes can adopt either a parallel topology, in which the backbone of each nucleic acid strand of the G-quadruplex run in the same direction, or an antiparallel topology in which the backbones run in opposing directions. All RNA quadruplex structures currently reported exhibit a parallel topology, with one recent exception, that is, the Spinach aptamer[36]. The use of CD spectroscopy to determine the topology of DNA G-quadruplexes is well established[37] although its use in studies of RNA G-quadruplexes can be problematic because CD spectra of RNA parallel G-quadruplexes are very similar to those of A-form structures. Nevertheless, being aware of this ambiguity, we used CD experiment mainly to exclude the possibility of antiparallel topology of RG4 quadruplex. The typical spectra of antiparallel quadruplexes have a negative band at 260 nm and positive band at 295 nm. The CD spectra recorded for 5′tiRNA$^{Ala}$ in the presence of K⁺, Na⁺, or Li⁺ ions displayed only a positive band at

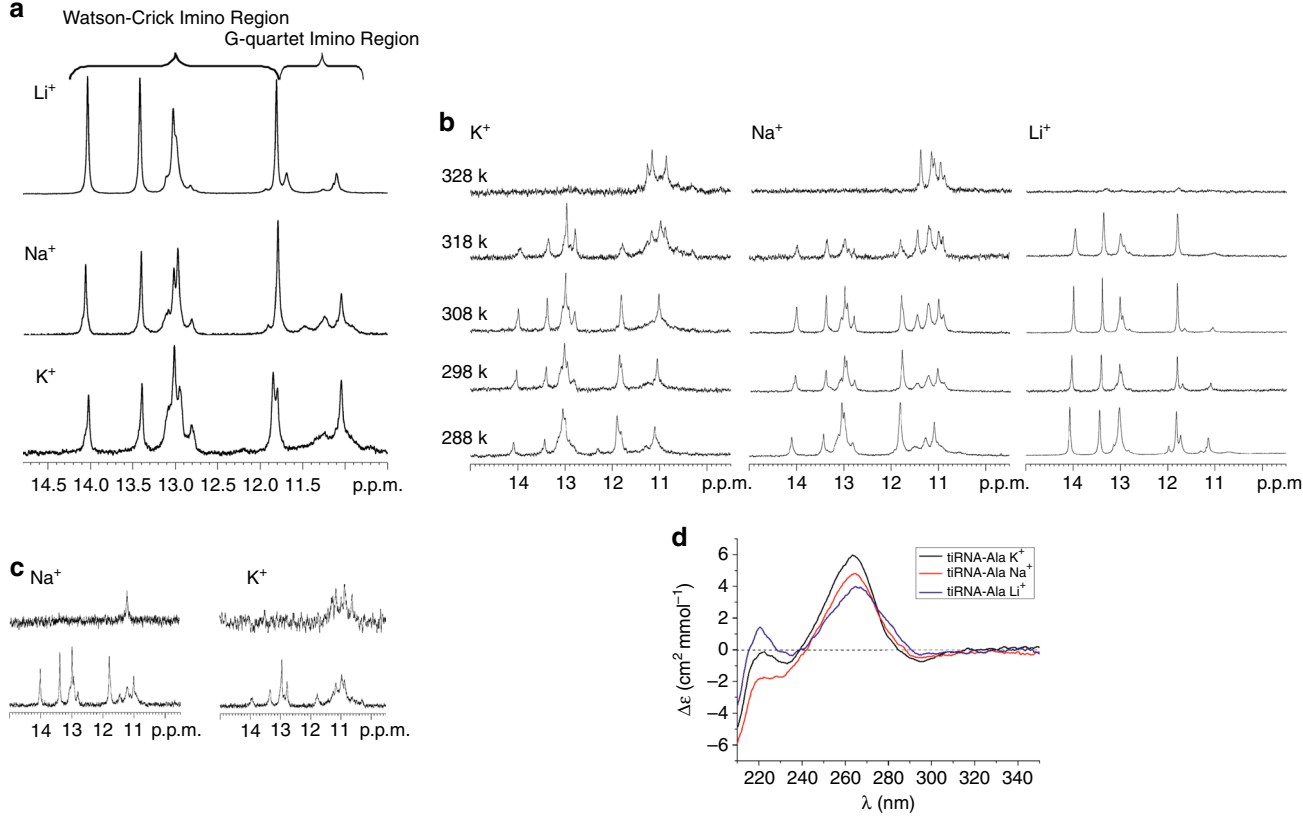

**Fig. 2** NMR spectra confirm G-quadruplex formation. **a** Imino region of the¹H NMR spectra of 5′tiRNA$^{Ala}$. Spectra were recorded at 25 °C in 90% H₂O/10% D₂O(v/v) in the presence of 50 mM KCl, 10 mM potassium phosphate, 0.1 mM EDTA, pH 6.8 (top panel), 150 mM NaCl, 10 mM sodium phosphate, 0.1 mM EDTA, pH 6.8 (middle panel), and 150 mM LiCl, 10 mM Tris-HCl, 0.1 mM EDTA, pH 6.8 (bottom panel). **b** Temperature dependence of the imino region of the ¹H NMR spectra of 5′tiRNA$^{Ala}$. Spectra were recorded in 90% H₂O/10% D₂O(v/v) in the presence of 50 mM KCl, 10 mM potassium phosphate, 0.1 mM EDTA, pH 6.8 (left panels), 150 mM NaCl, 10 mM sodium phosphate, 0.1 mM EDTA, pH 6.8 (middle panels), and 150 mM LiCl, 10 mM Tris-HCl, 0.1 mM EDTA, pH 6.8 (right panels). **c** Imino region of 5′tiRNA$^{Ala}$ in 90% H₂O/10% D₂O (lower panels) and after 12 h in D₂O (upper panels) at 45 °C in the presence of 150 mM NaCl, 10 mM sodium phosphate, 0.1 mM EDTA, pH 6.8 (left panels) and 50 mM KCl, 10 mM potassium phosphate and 0.1 mM EDTA, pH 6.8 (right panels). **d** CD spectra of 5′tiRNA$^{Ala}$ at 25 °C. The spectra were recorded in the presence of 50 mM KCl, 10 mM potassium phosphate and 0.1 mM EDTA, pH 6.8 (black), 150 mM NaCl, 10 mM sodium phosphate, 0.1 mM EDTA, pH 6.8 (red) and 150 mM LiCl, 10 mM Tris-HCl, 0.1 mM EDTA, pH 6.8 (blue)

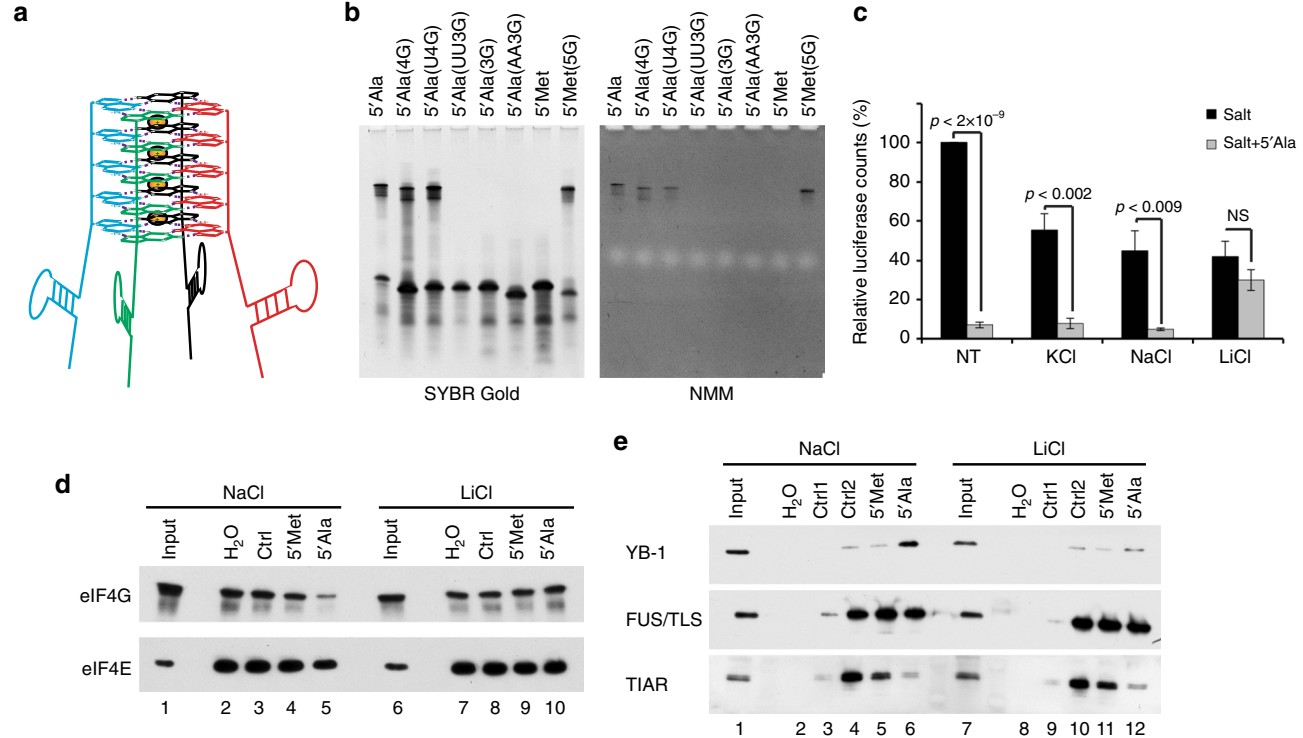

**Fig. 3** Deletion of RG4 structures in 5′tiRNA^Ala through ionic equilibration abolishes activity. **a** Model of tetrameric 5′tiRNA^Ala coordinated by a central G-quadruplex core containing 5 stacked G-quartets. **b** The 5′TOG motif of 5′tiRNA^Ala was mutated to varying extents. Indicated 5′tiRNA^Ala mutants were run on a 15% denaturing acrylamide gel and post-stained with SYBR gold or NMM to detect total RNA or RG4, respectively. The propensity to form RG4 structures correlated with the bioactivity of each tiRNA (Supplementary Table 2). **c** 5′tiRNA^Ala was equilibrated overnight in indicated salt or left untreated (NT). In vitro translation assays were carried out in rabbit reticulocyte lysate using polyadenylated NanoLuc RNA as a reporter. To each, 1 μl of 100 pmol salt equilibrated 5′tiRNA^Ala or 1 μl of indicated salt was added prior to initiating translation reaction. Depletion of RG4 structure of 5′tiRNA^Ala through equilibration in Li⁺ ions abolished the ability of 5′tiRNA^Ala to repress translation. **d** eIF4F was assembled onto m⁷GTP agarose and challenged with indicated RNAs equilibrated in NaCl or LiCl. Depletion of RG4 structures through equilibration in Li⁺ ions abrogated the ability of the tiRNA to displace eIF4G or eIF4E from m⁷GTP cap. **e** RNPs were purified from U2OS lysates using biotinylated RNAs equilibrated in indicated salt solutions. Depletion of RG4 structures reduced 5′tiRNA^Ala's association with YB-1, a key regulator of 5′tiRNA^Ala induced stress granules, to background levels. The interaction of 5′tiRNA^Ala with FUS/TLS or TIAR was unaltered through ionic equilibration

~ 265 nm and small negative band at 240 nm (Fig. 2d). No positive peak at 295 nm was detected implying that RG4 5′tiRNA^Ala adopts a parallel structure.

A typical G-quadruplex stem is formed from at least two stacked G-quartets. In a structure of RG4 5′tiRNA^Ala information regarding a number of stacked G-quartets was deduced from ¹H NMR spectrum recorded at 55 °C in solution containing Na⁺ ions (Fig. 2b). In this spectrum only five well resolved resonances were observed suggesting that 5′TOG motif is engaged in quadruplex formation. Although the number of RNA strands involved in RG4 formation could not be directly inferred from NMR spectra, however, gel retardation experiments demonstrated that the slowly migrating band of 5′tiRNA^Ala was a tetramer. Therefore, the only possible interpretation of the NMR data is that RG4 5′tiRNA^Ala adopts highly symmetric, parallel tetramolecular quadruplex with five G-quartets (Fig. 3a).

We present our model as for interacting 5′tiRNA^Ala strands; however, there is no apparent reason why exclusively homogeneous tetramer would predominant in a mixed population of other TOG-containing tiRNAs. We have presented data that 5′tiRNA^Cys also contains a TOG motif, but analysis of the genomic tRNA database (GtRNAdb, http://gtrnadb.ucsc.edu,[38]) reveals that certain tiRNAs derived from tRNA^Asp, tRNA^Val, and tRNA^Tyr also contain TOG motifs (Supplementary Table 1). Therefore, we hypothesized that heterogeneous tetramers could

form between different TOG-containing tiRNAs. To test this hypothesis, we equilibrated biotinylated tiRNA^Cys with unbiotinylated 3′ truncations of tiRNA^Ala (24mer and 27mer) (Supplementary Fig. 1b). The truncations served to distinguish between the tiRNAs on the basis of size without disrupting the TOG motif. tiRNA^Cys was recovered with streptavidin-agarose and RNAs that co-purified with it were eluted by exchanging NaCl with LiCl, thereby disrupting any RG4 structures. Eluted RNAs were analyzed on denaturing polyacrylamide gels stained with SYBR gold. Indeed, 5′tiRNA^Cys was able to interact with and co-purify 5′tiRNA^Ala molecules, where as non-TOG-containing biotinylated oligonucleotide, Ctrl1 did not purify 5′tiRNA^Ala (Supplementary Fig. 1c). These data suggest the possibility that heterogeneous TOG-containing tiRNA tetramers can form.

In summary, our data indicate that 5′tiRNA^Ala can adopt two conformations, an intramolecular hairpin structure (Fig. 1b) and tetrameric RG4 structure (Fig. 3a). In K⁺ and Na⁺ solutions equilibrium between hairpin and RG4 structures exists, however, only hairpin structure forms in the presence of Li⁺ ions.

**Importance of RG4 for 5′tiRNA^Ala biological activities.** The biological activity of 5′tiRNA^Ala resides within the 5′TOG motif

as judged by the abilities of tiRNA variants to trigger SG formation, displace eIF4F from $m^7GTP$ cap or inhibited translation of mRNA reporters[25]. While deletion of a single guanine (4G) or its substitution with a uracil (U4G) does not affect these activities, deletion or substitution of two or more guanines abrogated them (3G, UU3G) (Supplementary Table 2[25]). Conversely, 5′tiRNA$^{Met}$, which is biologically inert, can be "activated" upon addition of a 5′TOG motif (5′tiRNA$^{Met}$ (5G)). Analysis of these tiRNAs by SYBR gold or NMM on denaturing acrylamide gels demonstrates that only those tiRNAs which formed tetrameric RG4s retained biological activity (5′tiRNA$^{Ala}$ (WT, 4G, U4G), 5′tiRNA$^{Met}$(5G)) while inactivated tiRNAs have lost RG4s (5′tiRNA$^{Ala}$ (3G, UU3G, AA3G), 5′tiRNA$^{Met}$(WT)) (Fig. 3b). The loss of RG4 tetramer correlates with the loss of bioactivity.

The correlation between RG4 formation and biological activity with 5′tiRNA$^{Ala}$ is intriguing, but does not prove that RG4 is required for activity. In these cases, to prevent RG4 formation, we introduced changes in the 5′TOG-motif that may unpredictably change their structures. Therefore, we cannot distinguish between sequence dependent and structure-dependent effects. In order to manipulate the structure independently of making sequence changes, we took advantage of the ability of ions to stabilize or destabilize RG4s. 5′tiRNA$^{Ala}$ was equilibrated in RG4-permissive ($Na^+$ or $K^+$) or non-permissive ($Li^+$) ionic buffers overnight as we previously showed, dramatically altered the ability of 5′tiRNA$^{Ala}$ to assume a G4 conformation (Figs. 1e, 2). In vitro mRNA translation assays were performed with equilibrated tiRNAs by monitoring luciferase activity as a readout of translation efficiency. We found that 5′tiRNA$^{Ala}$ equilibrated in RG4 promoting ions ($K^+$ and $Na^+$) retained translation repression activity; however, 5′tiRNA$^{Ala}$ equilibrated in RG4 destabilizing $Li^+$ ions abolished translation repression activity (Fig. 3c).

5′tiRNA$^{Ala}$ inhibits translation via displacement of eIF4F from the $m^7GTP$ caps of mRNAs[25]. We analyzed whether this ability was altered in the ionic conditions affecting RG4 tetramers. We purified eIF4F complex on $m^7GTP$ agarose and challenged the resulting complexes with 5′tiRNA$^{Ala}$ equilibrated in NaCl or in LiCl. In accordance with translation repression data, when 5′tiRNA$^{Ala}$ cannot efficiently form RG4s, it is no longer able to displace eIF4F complex components eIF4E or eIF4G from $m^7GTP$ (Fig. 3d, Supplementary Fig. 2a, b). These data along with mutagenesis data (Fig. 3b, Supplementary Table 2) strongly suggest that RG4 structures within 5′tiRNA$^{Ala}$ are necessary for translation inhibition.

Overexpression of ANG induces expression of tiRNAs and potentiates cells toward the formation of SGs[29]. Further, transfection of cells with 5′tiRNA$^{Ala}$ triggers the formation of SGs. Interaction of 5′tiRNA$^{Ala}$ with Y-box binding protein 1 (YB-1) is absolutely required for formation of tiRNA-induced SGs[39]. We asked if RG4 structures were required for this interaction and therefore SG formation in response to 5′tiRNA$^{Ala}$. Cell lysates were prepared using buffers containing either NaCl or LiCl and 5′tiRNA-protein complexes were purified using biotinylated tiRNAs or control RNAs equilibrated in NaCl or LiCl. Resulting complexes were analyzed by western blotting (Fig. 3e). YB-1 readily interacted with RG4-containing 5′tiRNA$^{Ala}$ equilibrated in NaCl, but the interaction was reduced to background levels under RG4-non-permissive conditions. In contrast, FUS/TLS, a protein which interacts non-specifically with 5′tiRNA$^{Ala}$, was not altered and TIAR, a SG component, did not interact with 5′tiRNA$^{Ala}$ under either condition. This suggests that the YB-1:5′tiRNA$^{Ala}$ interaction may require RG4 structures and that SG formation may depend upon RG4s. However, due to the non-physiological levels of $Li^+$ ions, we were not able to directly test this hypothesis using this approach.

To overcome the challenges presented by non-physiological ions, we used the 5′tiRNA$^{Ala}$ variants bearing 7-deazaguanine (7daG), a guanine derivative, in the 5′TOG motif. Formation of RG4s via Hoogsteen base pairing strictly requires nitrogen at position 7 (N7) of a guanine, and its substitution to carbon atom in 7daG fails to support G-quartet formation (Fig. 4a)[40,41]. As RG4 formation requires adjacent G-quartets to stack upon each other, we reasoned that by disrupting Hoogsteen base pairing of adjacent guanosines, we might prevent RG4 formation. Therefore, we synthesized 5′tiRNA$^{Ala}$ oligonucleotides in which the guanines at position 2 and 4 were substituted for 7daGs (5′tiRNA$^{Ala}$ (7daG) TOG) (Fig. 4b), and their mobility in native and denaturative gels was compared with unmodified 5′tiRNA$^{Ala}$ under RG4-permissive or non-permissive ionic conditions. In contrast to 5′tiRNA$^{Ala}$, 5′tiRNA$^{Ala}$(7daG) was not able to form RG4 under any ionic conditions on both denaturing (Fig. 4c) or native (Fig. 4d) gels as monitored by SYBR Gold or NMM. Thus, 5′tiRNA$^{Ala}$ RG4 formation requires canonical Hoogsteen base pairing and presents an alternative method to test the reliance of RG4 formation on biological activity in the absence non-physiological ionic conditions.

We asked to what extent abolishing RG4 structures via 7daG substitution affected 5′tiRNA$^{Ala}$ biological activities under physiologically relevant conditions. First, we found that preventing RG4 formation relieved 5′tiRNA$^{Ala}$'s ability to repress translation >20-fold in an in vitro translation assay (Fig. 5a). Second, upon transfection into U2OS cells, 5′tiRNA$^{Ala}$(WT) readily induced the formation of SGs, while this ability was abolished in the non-G4 forming 5′tiRNA$^{Ala}$(7daG) (Fig. 5b). It is possible that the RG4 structure provides additional stability to 5′tiRNA$^{Ala}$(WT) that is lost in the 5′tiRNA$^{Ala}$(7daG) upon transfection into cells. To account for this possibility, we extracted RNA from cells post-transfection and assayed for the presence of transfected tiRNAs. We found no differences between 5′tiRNA$^{Ala}$(WT) and 5′tiRNA$^{Ala}$(7daG) (Supplementary Fig. 3a). This correlated with the failure of non-G4-forming 5′tiRNA$^{Ala}$ to interact with YB-1 (Fig. 5c) and displace eIF4E/eIF4G complexes from $m^7GTP$ agarose (Fig. 5d, Supplementary Fig. 3b). Non-RG4 forming 5′tiRNA$^{Ala}$(7daG) also failed to interact with known RG4 interacting proteins FXR1 and DHX36. Note that its interaction with Vigilin (VIG), Nucleolin (NCL), GRSF1, FUS/TLS, and TDP43 was unaltered suggesting that these factors interact in a sequence- (e.g., G-rich sequences) rather than a structure-dependent manner. Third, we asked whether 5′tiRNA$^{Ala}$(7daG) lost the ability to displace eIF4F components from the $m^7GTP$ cap. While 5′tiRNA$^{Ala}$(WT) readily displaced eIF4G and a portion of eIF4E, 5′tiRNA$^{Ala}$(7daG) or control RNAs were unable to do so (Fig. 5d, Supplementary Fig. 3c). These data, in combination with data generated using various ionic conditions, demonstrate that biological activity of 5′tiRNA$^{Ala}$ is dependent upon the RG4 structure formed by the 5′TOG motif. The sequence of the 5′TOG motif is only important in so much as it grants the ability to form this structure, not through sequence dependent interactions with proteins

**Tetramolecular RG4-tiRNAs are evolutionarily conserved.** tRNA cleavage in response to cellular stress is an evolutionary conserved phenomenon[42,43]. In *Arabadopsis thaliana*, tRNA fragments accumulate to high concentration and their abundance gives an indication to their potential importance[44]. Three of these fragments, At69B, At90, and At112A (derived from tRNA$^{Cys}$, tRNA$^{Ala}$, and tRNA$^{Leu}$, respectively) contain G-rich 5′ ends (Fig. 6a) and reported to inhibit translation of a luciferase reporter in vitro in a manner poorly understood mechanistically. We reasoned that they may inhibit translation through a RG4-

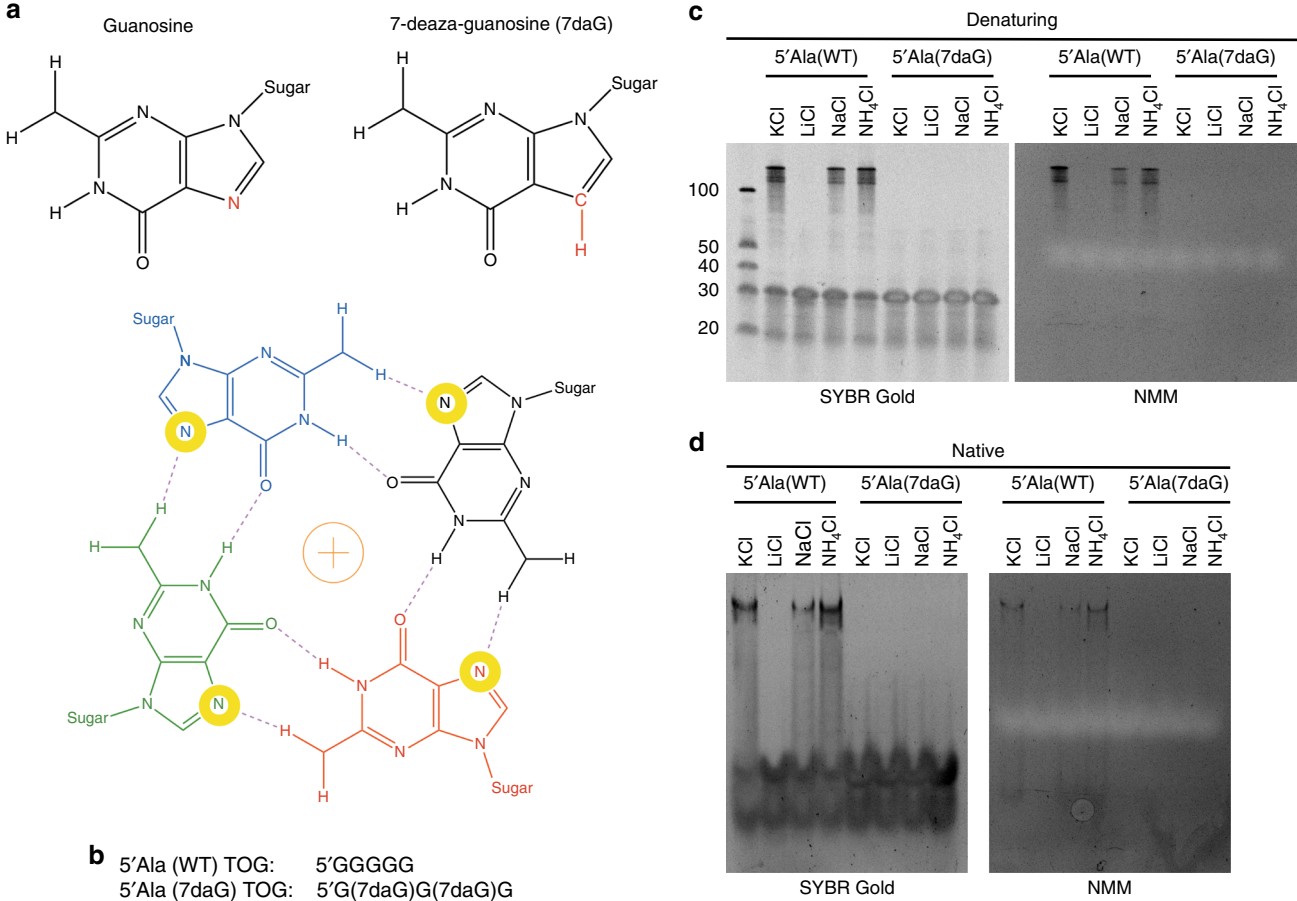

**Fig. 4** Substitution of guanines in 5′TOG motif with 7-deazaguanine prevents RG4 formation. **a** 7-deazaguanine derivative replaces the azide group at position 7 with a methine group (highlighted). Interaction with the hydrogen of the methine group is essential for Hoogsteen basepairing in a G-quartet. **b** Guanines at position 2 and 4 in the 5′TOG motif of 5′tiRNA$^{Ala}$ were substituted with 7-deazaguanine (**c**, **d**) 5′tiRNA$^{Ala}$(WT) or 5′tiRNA$^{Ala}$(7daG) were equilibrated overnight in indicated salt solution and analyzed on denaturing **c** or native **d** gel and post-stained with SYBR gold (left panels) to detect total RNA or NMM (right panels) to detect RG4 structures. Regardless of ionic condition, 5′tiRNA$^{Ala}$(7daG) failed to form RG4 structure highlighting the importance of Hoogsteen base pairing for RG4 formation

dependent mechanism as human 5′tiRNA$^{Ala}$. We first analyzed the mobility of these tRNA fragments equilibrated in either Na$^+$ or Li$^+$ on denaturing gels stained with SYBR Gold or NMM. At90 readily formed a more slowly migrating RNA species consistent with tetrameric RG4 structures that were abrogated in the presence of Li$^+$ cations (Fig. 6b). The other two RNA species either failed to form RG4 structures (At68B) or only an extremely minor fraction formed RG4 structure (At112A), correlating with the number of 5′ guanines (3 vs. 4).

We next tested if RG4 structures are responsible for translation repression activity of these RNAs. Each small RNA was equilibrated in RG4-permissive salts (NaCl) or RG4-non-permissive salts (LiCl). Translation repression activity was analyzed by their ability to repress a luciferase reporter. Without salt equilibration, we recapitulated previously reported data in that each tRNA fragment inhibited translation (Fig. 6c, NT). The ability of At90 RNA to repress translation was markedly reduced by treatment with LiCl (Fig. 6c), which correlated with destabilization of RG4 structures (Fig. 6b). This mirrors the results found with human 5′tiRNA$^{Ala}$ (Figs. 3, 5), suggesting evolutionary conservation of the utilization of RG4-containing small RNAs to inhibit translation. However, regardless of ionic conditions, At68B and At112A retained their translation inhibitory activities, suggesting they utilize an alternative, non-

RG4 pathway to inhibit translation. Nonetheless, these data also suggest that translation repression mechanisms relying on the assembly of intermolecular RG4 structures are present in distinct phyla.

## Discussion

Here we show that 5′tiRNA$^{Ala}$ assemble into tetrameric G-quadruplexes mediated by the 5′TOG motif. Through analysis of 5′tiRNA$^{Ala}$ by NMR, circular dichroism, gel retardation and affinity to RG4-ligand NMM, we have developed a structural model of 5′tiRNA$^{Ala}$ (Fig. 3a). Monomeric 5′tiRNA$^{Ala}$ is capable of forming a hairpin in which the 5′ end, consisting of the 5′TOG motif, remains unpaired (Fig. 1b). Via Hoogsteen base pairing, guanines in the 5′TOG motif can interact with other monomeric 5′tiRNA$^{Ala}$ forming G-quartets (Figs. 1a, 3a). This structure represents a new class of RNA G-quadruplexes in which stretch of five consecutive guanines is engaged in RG4 formation. Although, it is worth recognizing that these structures have been proposed to exist for DNA G4[45]. The number of known RNA G-quadruplex structures is still very limited[12] and large number of diverse RG4 topologies is still unknown, waiting to be explored. This structure represents an endogenous intermolecular RNA G-quadruplex.

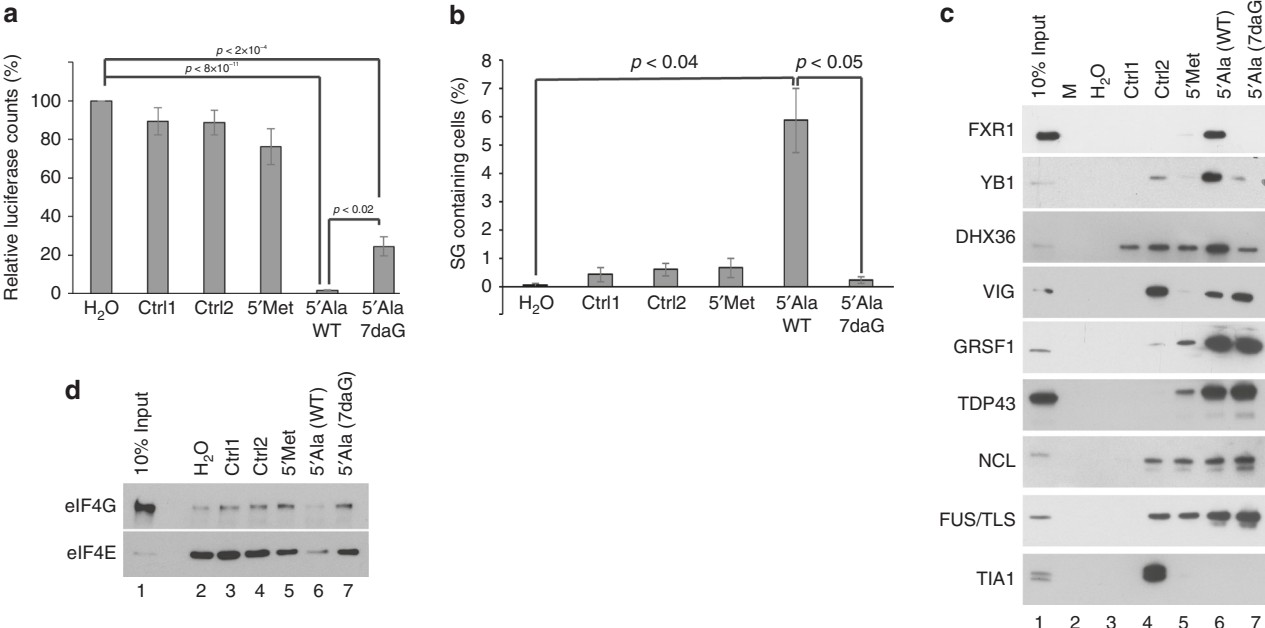

**Fig. 5** 7-deazaguanine substitutions in 5′tiRNA[Ala] prevent bioactivity. **a** In vitro translation assay in rabbit reticulocyte lysate using NanoLuc reporter to monitor translation efficiency. Substitution of two guanines in 5′tiRNA[Ala] reduces its ability to repress translation. **b** Indicated RNAs were transfected into U2OS cells and stress granule formation was monitored through immunofluorescence. 5′tiRNA[Ala] (WT) readily induces the formation of SGs, while 5′tiRNA[Ala](7daG) fails to. **c** RNA affinity purifications using indicated RNAs from U2OS lysates. Loss of RG4 forming ability in 5′tiRNA[Ala] (7daG) correlates with loss of binding to YB-1, a protein required for tiRNA mediated SG formation. 5′tiRNA[Ala](7daG) also loses the ability to bind to Fxr1 and DHX36, previously identified RG4 binding proteins. In contrast, Vigilin, GRSF1, and TDP43 bind 5′tiRNA[Ala] regardless of its ability to form RG4. **d** Inhibition of RG4 formation in 5′tiRNA[Ala] through incorporation of 7-deazaguanine abrogates its ability to displace eIF4F from m[7]GTP agarose

Previous mutational analysis of 5′tiRNA[Ala] showed that guanosines present in the 5′TOG motif are required for activity[25]. By introducing guanine to uracil transversions, we abolished activity of 5′tiRNA[Ala] and speculated that it was due to the loss of interaction with a vital protein co-factor[25,30]. Through manipulation of ionic conditions to prevent RG4 formation or by substitution of nitrogen atom with carbon at the position 7 of guanosine, we have shown that the bioactivity of 5′tiRNA[Ala] requires tetrameric RG4 formation. Thus, 5′tiRNA[Ala] inhibits translation through displacement of eIF4F from the m[7]GTP cap of mRNA thereby triggering the formation of stress granules and abrogation of RG4 formation prevents these activities.

The use of the 7-deazaguanine is revealed to be a powerful tool in probing the relative contribution of Watson–Crick and Hoogsteen interactions in a biological action[40,41]. Using 7daG modified oligonucleotides, we were able to clearly distinguish between interactions which required Watson–Crick interactions and those that required Hoogsteen interactions. By combining this analysis with ionic equilibrations that prevented RG4 assembly, we can clearly distinguish between sequence specific and structure specific interactions (Fig. 5c). For example, YB-1, a protein required for 5′tiRNA[Ala] induced stress granules interacts only with the RG4 form of 5′tiRNA[Ala], while GRSF1, a protein with no known function in 5′tiRNA[Ala] bioactivity, interacts in a sequence dependent manner. The use of this strategy to determine the role of G-quadruplexes in biology holds great promise that is only begin to be realized.

Our initial mutagenesis also revealed that nucleotides located within the loop of the stem-loop play an additive role in activity. This raises the intriguing possibility that the RG4 structure serves as a mechanism to concentrate factors by acting as a molecular scaffold. A tetrameric 5′tiRNA[Ala] brings four stem-loop structures into proximity. Through interaction with key proteins, a high-

local concentration would be achieved. This is particularly exciting in the context of stress granule formation[28,46]. Stress granules form through phase transitions that are highly dependent upon high-local concentrations of certain key protein factors (G3BP1/2, TIA1/TIAR, and so on)[47]. Canonical SG formation requires activation of one of four stress activated kinases that phosphorylates the alpha subunit of eIF2 to inhibit translation initiation[27]. However, 5′tiRNA induced SGs are independent of eIF2α phosphorylation, while dependent on the activity of RNA-binding protein YB-1[39]. We propose that concentration of YB-1 through aggregation of tetrameric 5′tiRNA[Ala] may provide a mechanism for eIF2α phosphorylation independent SG formation (Supplementary Fig. 4).

Recently, questions have been raised about the prevalence of RG4s in eukaryotic cells. It has been argued that, despite immunofluorescence and functional data from other labs, in human and yeast cells RG4s are actively unwound[18]. It is possible that during mRNA biogenesis, specific factors bind mRNAs co-transcriptionally and keep RG4s unfolded. Upon export of transcript into cytoplasm, these factors help the ribosome to initiate on mRNAs by preventing assembly of RG4-roadblocks. In contrast to mRNAs that have a nuclear prehistory, RG4 assembly by tiRNAs is unique because it is dependent on cleavage of cytoplasmic tRNAs, molecules that are immediately available upon cellular stress. Therefore, we believe that investigation of RG4 formation during transient cell stress represents a promising and understudied field.

## Methods

**NMR spectroscopy**. NMR spectra were collected on a Bruker AVANCE III 700 MHz spectrometer equipped with a QCI CryoProbe. The 5′tiRNA[Ala] samples were dissolved in buffers containing (i) 150 mM NaCl, 10 mM Na$_2$HPO$_4$/NaH$_2$PO$_4$, 0.1 mM EDTA, (ii) 50 mM KCl, 10 mM K$_2$HPO$_4$/KH$_2$PO$_4$, 0.1 mM

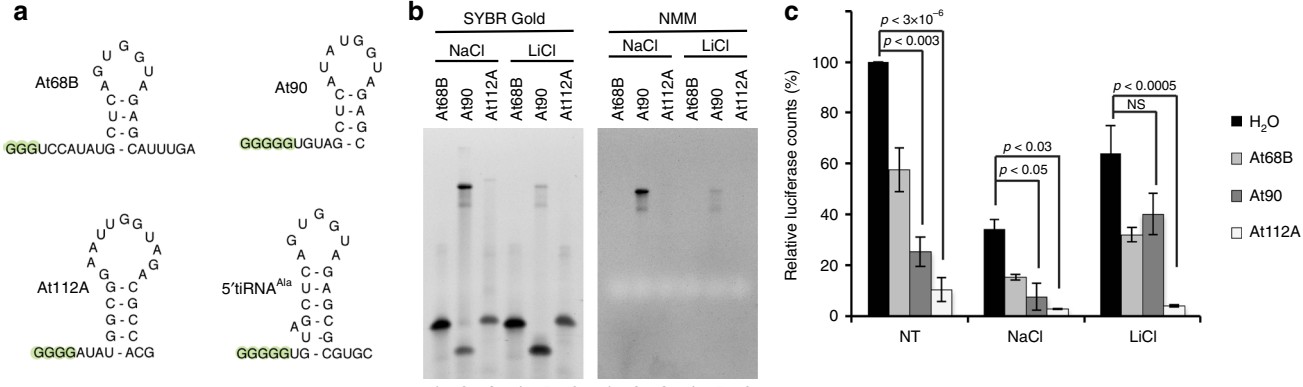

**Fig. 6** tRNA fragments from *Arabadopsis thaliana* form RG4 which are required for their activity. **a** Secondary structures of three 5′tiRNA species previously identified in salt stressed *A. thaliana* harboring putative 5′TOG motifs. **b** Analysis of *A. thaliana* tiRNAs on 15% denaturing acrylamide gel post-stained with either SYBR Gold or NMM to detect total RNA or RG4 structures, respectively. AtRNA90 forms a strong RG4 positive band which runs in a similar manner to 5′tiRNA$^{Ala}$. Upon equilibration in LiCl, the more slowly migrating RG4 positive band disappears. **c** In vitro translation in rabbit reticulocyte lysates using salt equilibrated or untreated (NT) *A. thaliana* RNAs. Depletion of RG4 structures in AtRNA90 inhibits its ability to repress translation. However, other active *A. thaliana* tiRNAs repress translations despite ionic equilibration suggesting they inhibit translation via an alternative mechanism

EDTA, or (iii) 150 mM LiCl, 10 mM Tris-HCl, 0.1 mM EDTA (pH 6.8). The 3 mm thin wall tubes were used with a final sample volume of 200 µl. The 5′tiRNA$^{Ala}$ samples were annealed by heating to 90 °C and then slowly cooled down to the room temperature. It was verified that the relative ratio of signals in NMR spectra did not change over a few weeks. The final concentration of 5′tiRNA$^{Ala}$ was 0.17 mM. A mixture of 90% $H_2O$ and 10% $D_2O$ was used for experiments undertaken to study exchangeable protons. The water signal was suppressed by excitation sculpting with gradient pulse[48]. The 2D NOESY spectrum was collected from 1840 scans in a buffer containing Na$^+$ cations at 10 °C. For experiments carried out in $D_2O$, the oligonucleotide was dried and redissolved in 99.996% $D_2O$ and $D_2O$ residual water signal was suppressed using the low power presaturation. Spectra were processed and prepared with TopSpin 3.0 Bruker Software.

**Circular dichroism.** The JASCO J815 spectropolarimeter was used to collect CD spectra. The oligonucleotides were dissolved in the same buffers as for NMR studies. Quartz cuvettes with a path length 0.5 cm were used with the sample volumes of 1300 µl to achieve a sample concentration of 4.3 µM. Spectra were collected in the range between 220 and 340 nm at 25 °C from three scans and a buffer baseline was subtracted from each sample. CD was expressed as the difference in the molar absorption of the right-handed and left-handed circularly polarized light. $\Delta\varepsilon$, in units of cm$^2$ mmol$^{-1}$, was normalized to the number of nucleoside residues in the RNA samples[49].

**Salt equilibration and staining.** RNA oligonucleotides were synthesized by Integrated DNA Technologies (IDT). To equilibrate, RNAs were diluted to 10 µM in indicated salt solution, heated to 95 °C for 10 min and allowed to cool to room temperature overnight. For analysis on acrylamide gels, 10 pmol or 50 pmol were ran through a gel and stained with SYBR gold or NMM, respectively. For SYBR gold staining, gels were post-stained in a solution of 1X SYBR Gold (ThermoFisher Scientific) in 0.5X TBE for 10 min. For NMM staining, gels were post-stained in a solution 0.1 mg ml$^{-1}$ of NMM in 0.5X TBE for 10 min. Following staining, gels were destained for 20 min in 0.5X TBE while rocking at room temperature. Gels were visualized using a 265 nm UV transilluminator.

**In vitro translation assay.** Nanoluciferase (Promega) was subcloned from pNL1.1 into pT7-5′/3′MCS-A50 vector generated in house. pT7-NLuc-A50 was linearized with SacII and in vitro transcribed using T7 RiboMAX Express Large Scale RNA production System (Promega). Transcribed reporter RNAs were extracted with acid equilibrated phenol/chloroform and ethanol precipitated with 2.5 M ammonium acetate. Resulting RNAs were resuspended in DEPC-treated water and purified through Illustra MicroSpin G-25 column (GE Life Sciences). The resulting RNA was uncapped and contained a 50 nt synthetic poly(A) tail. In vitro translation assays were carried out in rabbit reticulocyte lysate system (Promega) following established protocols using 100 ng of in vitro transcribed RNA and 100 pmol of indicated tiRNA or control RNA. Reactions were incubated at 30 °C for 30 min. Luciferase activity was assayed using Nanoluc Luciferase assay Kit (Promega) on Luminometer (GloMax Explorer, Promega) according to manufacturer's instructions (0.3 s measurement time).

**m$^7$GTP binding assay.** Cell lysates were prepared from U2OS cells (ATCC, USA) by lysis with NP-50 Lysis buffer (10 mM Tris (pH 8.0), 100 mM NaCl, 0.5% NP-40) and eIF4F was assembled onto m$^7$GTP agarose (Jena Bioscience) by tumbling for 2 h. Assembled complexes were further purified by washing with NP-40 lysis buffer to remove unbound proteins[25]. Assembled eIF4F complexes were challenged with 100 pmol of indicated RNA for 2 h at 4 °C while tumbling. Displaced proteins were washed away with NP-40 Lysis buffer and bound proteins were eluted with 60 µL of 1X SDS Loading dye. eIF4F displacement was assessed by analyzing eIF4G and eIF4E by western blotting.

**Stress granule induction and quantification.** Cells were grown on glass coverslips in 24-well dish until ~75% confluent. Cell were transfected with 250 pmol of indicated RNA. 6 h post-transfection, cells were fixed with 4% paraformaldyhde in PBS for 15 min and permeabilized with ice-cold 100% methanol for 10 min and processed for fluorescence microscopy[29]. To ensure tiRNA stability, RNA was extracted from parallel transfected wells with TRIZOL. 10 µg of RNA was run on 15% denaturing acrylamide gels, SYBR gold stained and transferred to Hybond N$^+$ nylon membrane. Blots were probed with streptavidin-HRP to detect biotinylated RNAs. Stress granules were monitored by co-localization of foci positive for G3BP1, eIF4G, and TIAR according to established protocols[50–52]. Only cells with granules co-staining for these markers were considered SGs and a minimum of three SGs per cell were required to score positive. Coverslips were mounted in polyvinyl mounting media and viewed at room temperature using a Nikon Eclipse E800 microscope with a 40X Plan Fluor (NA 0.75) or 100X Plan Apo objective lens (NA 1.4) and illuminated with a mercury lamp and standard filters for DAPI (UV-2A–360/40; 420/LP), Cy2 (FITC HQ 480/40; 535/50), Cy3 (Cy 3HQ 545/30; 610/75), and Cy5 (Cy 5 HQ 620/60; 700/75). Images were captured with SPOT Persuit digital camera (Diagnostic Instruments) with the manufacturers software and compiled using Adobe Photoshop CC 2015.

**RNA affinity chromatography.** tiRNA binding proteins were affinity purified as previously described[25,39]. Briefly, U2OS cells were grown to ~75% confluence in a 15 cm tissue culture dish and lysed with NP-40 Lysis buffer and insoluble material was removed following centrifugation. Supernatant was split into individual tubes to which 250 pmol of indicated RNA oligonucleotides were added. Oligonucleotides contained a biotin moiety at their 3′ end. Oligonucleotides were tumbled with lysate for 2 h at which point 10 µL of streptavidin-agarose was added and tumbled for an additional 2 h at 4 °C. Unbound proteins were washed away with NP-40 Lysis buffer and bound proteins were recovered by eluting with 60 µL of 1X SDS loading buffer.

**Heterogeneous tiRNA tetramer capture.** 100 pmol of biotinylated RNA (5′tiRNA$^{Cys}$ or Ctrl1) was equilibrated with 300 pmol of unbiotinylated 5′tiRNA$^{Ala}$ (24mer) or 5′tiRNA$^{Ala}$(27mer) in 150 mM NaCl. Biotinylated complexes were recovered with streptavidin-agarose by tumbling at 4 °C for 2 h. Complexes on beads were washed 1X with 150 mM NaCl, 1X with dH$_2$O, and 1X with 150 mM LiCl before resuspending in 50 µl of 150 mM LiCl and heating to 100 °C and allowing to cool to room temperature. Eluted supernatants were analyzed on 15% Urea-acrylamide gel and visualized by SYBR Gold.

**Oligonucleotides**. All oligonucleotides were synthesized and purified by IDT. Their sequences can be found in Supplementary Table 3.

**Antibodies**. Full table and dilutions for each antibody used can be found in Supplementary Table 4. YB-1 (Abcam: Ab12148), G3BP1 (Santa Cruz: sc81940), TIA1 (Santa Cruz: sc-1751), TIAR (Santa Cruz: sc-1749), eIF4G (Santa Cruz: sc-11373), Nucleolin (Santa Cruz: sc-9893), eIF4E (Santa Cruz: sc9976), Fus/TLS (Protein Tech Group: 11570-1-AP), TDP43 (Protein Tech Group: 10782-2-AP), Vigilin (Santa Cruz: 2404C3a), DHX36 (Protein Tech Group: 13159-1-AP), and GRSF1 (Aviva: ARP40382-P050).

**Data availability**. The data that support the findings of this study are available from the corresponding author upon request.

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

## Acknowledgements

We thank members of the Ivanov and Anderson labs for helpful discussion and feedback on this manuscript. This work is supported by the National Institutes of Health

(NS094918 to P.I., GM121410 to P.I. and Paul Anderson, F32 GM119283 to S.M.L.) and the National Science Center (UMO-2014/13/B/ST5/04144 to Z.G.).

## Author contributions

D.G. and Z.G. designed and undertook the biophysical experiments and analyzed the data. S.M.L. and S.M.C. performed functional studies and analyzed the data. P.I. conceived the project, supervised project, and analyzed the data. P.I., S.M.L., and Z.G. wrote the manuscript.

## Additional information

**Competing interests:** The authors declare no competing financial interests.

**Change history:** A correction to this article has been published and is linked from the HTML version of this paper.

