## [Peer Review file · Nature Communications]

Reviewers' comments:

Reviewer #1 (Remarks to the Author):

While tRNA-derived RNA fragments have been first described already in the late 1970-ies and are continuously detected in basically every RNA deep sequencing project, functional characterization of this ncRNA class remained sparse. The submission by Lyons et al. is one of the few positive exceptions since it clearly focuses on gaining functional insight into tRNA fragment biology which goes beyond a pure descriptive dimension. By cleverly combining different experimental approaches embracing biophysical, biochemical as well as molecular cell biological techniques the authors provide evidence, that a particular human 5'-tRNA half forms intermolecular RNA G-quadruplex (RG4) structures that inhibit translation by sequestering eIF4F from the mRNA cap and triggers the formation of stress granules. Even though the available data do not yet allow drawing a complete mechanistic in vivo model of this tRNA half molecule, it represents a major advance in the ncRNA field and thus is of general interest.

Minor points:

- 1) Abstract, line 41: Since the authors do not describe in vivo functions of the endogenous 5' tRNA-Ala half, the statement 'is absolutely required for functions of tRNA fragments in regulating mRNA translation and stress responses' is too strong and should be re-phrased and softened.
- 2) Page 3, lines 89-91: This sentence in its current form is unclear and needs to be rephrased.
- 3) Results chapter, 1st heading: The phrase 'Translationally-active' sounds a bit odd. In fact this paragraph does not describe any experiment that deals with translation.
- 4) At several places throughout the manuscript the characters of figures should be in lower case (e.g. lines 167, 229 and in all figure legends)
- 5) The existence of RG4 structures is most convincingly demonstrated by employing synthetic 5' tRNA-Ala half carrying 7-deaza-G substitutions at positions 2 and 4 of the 5' stretch of five guanosines (Figs 4 & 5). The rationale of replacing G2 and G4 and not any of the other 5' guanosines needs to be better explained.
- 6) Figs 3c and 6c: the abbreviation NT needs to be defined in the corresponding figure legends.
- 7) Line 271: it should read (Figure 5d & S3b) instead of (Figure 4d & S3b)
- 8) In order to be able to interpret the in vitro translation data shown in Figs. 3c and 6c the authors need to inform the reader, whether or not the tRNA halves were still intact after the overnight incubation in the different monovalent ion buffers. From the methods section it is not clear whether these overnight equilibrations have been performed as described for the gel retardation assays (lines 395-402) or not.
- 9) When looking at the data shown in Fig. 6c the sentence 'The ability of At90 RNA to repress translation was abolished...' (lines 293-294) is not justified. I agree that the translation inhibition of At90 is markedly diminished after LiCl equilibration, but by no means is it abolished.
- 10) Since the 5' tRNAAla half is not the only tRF produced upon stress that can inhibit translation by the mechanism described herein (e.g. 5'tRNA-Cys half; ref. 21), have the authors ever thought about the possibility that in vivo different tRFs form a heterogenic intermolecular RG4 structure that might represent the functional complex?

Reviewer #2 (Remarks to the Author):

The manuscript by Lyons et al investigates the contribution of RNA G-quadruplex formed in tRNA fragments in translation inhibition.

This articles follows from two publication by the same groups that i) showed that tiRNAs containing an 5' oligoguanine repress translation (Mol Cell, 2011) and ii) that these tiRNA 5' oligoguanine regions are able to adopt a G-quadruplex structure (PNAS, 2014).

In this manuscript, the authors nicely combined various methods to further characterize the contribution of these G-quadruplexes in translation repression and stress granule formation.

The manuscript is clearly presented and convincingly demonstrate that the G-quadruplexes formed in these tiRNAs are crucial for translation repression and stress granule formation.

The authors should consider the following points:

1- Lanes 189-192: The authors are correct that the CD spectrum observed correspond to a parallel G4. However, stem-loop RNAs display a very similar CD signature. Since the tiRNA -Ala also adopts a stem-loop structure (as demonstrated by the NMR data), the CD spectra can not be unambiguously attributed to a parallel RNA G4. However, the NMR data presented in Figure 2a-c are convincing.

2- Lanes 261-265: I was surprised to read that tiRNAs (natural and deaza) can be efficiently transfected into cells. This is very important and suggests that 7-deaza-modified RNAs can be investigated directly in cells, opening new opportunities of studying G4s in cells. I wonder how the authors confirmed that these RNAs were indeed successfully transfected into cells. The results corresponding to 5'Ala-7daG RNA presented in Figure 5b could also be interpreted by a failure to transfect the modified RNA in cells.

3- Lanes 297-299, Figure 6c: I do not agree with the authors that At68B has no effect on translation. The figure suggests that At68B behaves in a very similar manner as At90.

4- Lanes 313-320: The authors refer extensively to their previous article published in Mol Cell (2011) but less so to their previous article in PNAS (2014). The PNAS paper already demonstrated that tiRNAs can adopt G4 structures and that these structures were crucial for translation inhibition. It seems to me that this manuscript is a follow up study of their PNAS paper and therefore results from the PNAS paper should be discussed in this manuscript.

Minor points:

1- Lane 61: The sentence refers to DNA G4s but Reference 9 refers to RNA G4s. Reference 9 should be changed to Biffi et al, Nat. Chem., 2013.

2- Lane 265: Figure 5d should not be cited here.

3- Lane 271: Figure 5d should be cited in place of Figure 4d

Reviewer #3 (Remarks to the Author):

In this manuscript, Lyons et al. characterize an RNA G-quadruplex formed from tRNA fragments. The work focuses on 5'tiRNAALA which the authors purport forms a tetramolecular G4 and poses regulatory functions. This manuscript shows that tiRNA is able to form intermolecular G4 in vitro, and proposes a structure model for the inter-RNA-G4. A combination of gel-based and biophysical experiments were conducted. However, although the idea is interesting, similar results were published in their 2014 PNAS, with some data shown in the supporting information of PNAS. I am not sure whether this manuscript contains enough new information that warrants publication in Nature Comm.

Using in vitro translation and in vivo reporter assay, the authors show that this inter-RNA-G4 can represses mRNA translation by displacing eIF4 complex from the mRNA 5'-capping structure. It

would be interesting to understand how and why tetra-G4 can compete with the original substrate of eIF4.

Ctrl1 and Ctrl2 in figure 5c seem quite different. The author should provide detailed sequences for these controls.

As the work is more pertinent towards the how formation of tetramolecular G-quadruplex structure in the 5'tiRNA Ala sequence, authors should have explained more details about the structure of G-quadruplex formed. How do the authors determine which guanines are involved in the G4 formation? How is the parallel G4 structure model derived? How were assignments achieved in Figure S1?

The 5'tiRNA sequence mentioned in the Fig. 2b and Fig. 6a are different, which one is the correct one?

What are the full-length sequences of the tiRNAs used in this study?

The selected tRNA sequence has both stem-loop region and extended 5'-guanine rich sequence, how this will be shown in CD experiments was not explained. If the four tRNA molecules form tetramolecular structure, what happen to the stem and loop region?

The section title "Tetramolecular RG4-tiRNAs are present and functioning in plants" is misleading, as it may imply an in-cell or in-organism experiment.

Reviewer #1 (Remarks to the Author):

While tRNA-derived RNA fragments have been first described already in the late 1970-ies and are continuously detected in basically every RNA deep sequencing project, functional characterization of this ncRNA class remained sparse. The submission by Lyons et al. is one of the few positive exceptions since it clearly focuses on gaining functional insight into tRNA fragment biology which goes beyond a pure descriptive dimension. By cleverly combining different experimental approaches embracing biophysical, biochemical as well as molecular cell biological techniques the authors provide evidence, that a particular human 5'-tRNA half forms intermolecular RNA G-quadruplex (RG4) structures that inhibit translation by sequestering eIF4F from the mRNA cap and triggers the formation of stress granules. Even though the available data do not yet allow drawing a complete mechanistic *in vivo* model of this tRNA half molecule, it represents a major advance in the ncRNA field and thus is of general interest.

We thank this reviewer for positive evaluation of our work and suggestions for changes in the text.

Minor points:

1) Abstract, line 41: Since the authors do not describe *in vivo* functions of the endogenous 5'tRNA-Ala half, the statement 'is absolutely required for functions of tRNA fragments in regulating mRNA translation and stress responses' is too strong and should be re-phrased and softened.

We agree with the assessment that this statement is too strong and have re-phrased. We would argue that SG formation (Figure 5B) is an *in vivo* readout of biological activity. However, as this is only a minor aspect of our investigation, we agree that re-wording is appropriate.

Original Sentence:

RG4 is absolutely required for functions of tRNA fragments in the regulation of mRNA translation and stress responses, and its disruption abrogates their bioactivities both *in vitro* and *in vivo*.

Revised Sentence:

RG4 is required for functions of tRNA fragments in the regulation of mRNA translation, a critical component of cellular stress response. RG4 disruption abrogates tRNA fragments ability to trigger the formation of SGs *in vivo*.

2) Page 3, lines 89-91: This sentence in its current form is unclear and needs to be rephrased.

Original Sentence:

Here, we demonstrate that the 5'TOG motif promotes formation of an tetramolecular RG4 and that this structure, rather than simply the sequence is required for activity of 5'TOG containing tiRNAs.

Revised Sentence:

Here, we demonstrate the functional requirement for the 5'TOG motif. The stretch of 5 guanosine residues within the 5'TOG interacts with the 5'TOG motifs of other tiRNAs, forming tetramolecular RG4. We demonstrate that this structure is required for 5'tiRNA activity, rather than simply RNA-protein interactions driven by sequence determinants.

3) Results chapter, 1st heading: The phrase 'Translationally-active' sounds a bit odd. In fact this paragraph does not describe any experiment that deals with translation.

We agree!

Original Heading:

Translationally-active 5'tiRNA^{Ala} assembles two alternative structural forms

Revised Heading:

5'tiRNA^{Ala} assembles two alternative structural forms

4) At several places throughout the manuscript the characters of figures should be in lower case (e.g. lines 167, 229 and in all figure legends)

We have changed all references to figures to lower cases.

5) The existence of RG4 structures is most convincingly demonstrated by employing synthetic 5'tRNA-Ala half carrying 7-deaza-G substitutions at positions 2 and 4 of the 5' stretch of five guanosines (Figs 4 & 5). The rationale of replacing G2 and G4 and not any of the other 5' guanosines needs to be better explained.

Efficient RG4 formation requires stacking of adjacent G-quartets. By disrupting the Hoogsteen base pairing of alternate guanosines (2 & 4) and therefore, G-quartet formation, we reasoned that we could prevent assembly of the remaining G-quartets into a RG4. Ideally this could have also been accomplished by incorporation of 7-daG at positions 1, 3 and 5. However, the 7-daG modification is costly to incorporate into a synthetic nucleic acid, so, we chose to only incorporate at 2 positions rather than 3.

New Text:

As RG4 formation requires adjacent G-quartets to stack upon each other, we reasoned that by disrupting Hoogsteen base pairing of adjacent guanosines, we might prevent RG4 formation.

6) Figs 3c and 6c: the abbreviation NT needs to be defined in the corresponding figure legends.

NT refers to "No treatment". That is, they remain in the ionic conditions in which they were synthesized. This information has been added.

New Text:

Figure 3c

5'tRNA^{Ala} was equilibrated overnight in indicated salt or left untreated (NT).

Figure 6c

In vitro translation in rabbit reticulocyte lysates using salt equilibrated or untreated (NT) *A. thaliana* RNAs.

7) Line 271: it should read (Figure 5d & S3b) instead of (Figure 4d & S3b)

The text has been altered referencing the proper figure.

Revised Text:

While 5'tiRNA^{Ala}(WT) readily displaced eIF4G and a portion of eIF4E, 5'tiRNA^{Ala}(2x-7daG) or control RNAs were unable to do so (**Figure 5d & S3c**).

8) In order to be able to interpret the *in vitro* translation data shown in Figs. 3c and 6c the authors need to inform the reader, whether or not the tRNA halves were still intact after the overnight incubation in the different monovalent ion buffers. From the methods section it is not clear whether these overnight equilibrations have been performed as described for the gel retardation assays (lines 395-402) or not.

We agree that this was unclear in the original text. However, the reviewer is correct in assuming that they were done as previously described for the gel retardation assays. We have altered the text to clarify this point.

Original Text:

5'tiRNA^{Ala} was equilibrated in RG4-permissive (Na⁺ or K⁺) or non-permissive (Li⁺) ionic buffers and *in vitro* mRNA translation assays were performed with equilibrated tiRNAs by monitoring luciferase activity as a readout of translation efficiency.

Revised Text:

In order to manipulate the structure independently of making sequence changes, we took advantage of the ability of ions to stabilize or destabilize RG4s. 5'tiRNA^{Ala} was equilibrated in RG4-permissive (Na⁺ or K⁺) or non-permissive (Li⁺) ionic buffers overnight as we previously showed, dramatically altered the ability of 5'tiRNA^{Ala} to assume a G4 conformation (**Figure 1e and 2**). *In vitro* mRNA

translation assays were performed with equilibrated tRNAs by monitoring luciferase activity as a readout of translation efficiency.

9) When looking at the data shown in Fig. 6c the sentence ‘The ability of At90 RNA to repress translation was abolished...’ (lines 293-294) is not justified. I agree that the translation inhibition of At90 is markedly diminished after LiCl equilibration, but by no means is it abolished.

We agree with the reviewers assessment of the results. We originally used the phrase “abolished” in reference to the loss of statistical significance for the translation repression. However, there is still a non-statistically significant reduction, so the term “abolished” is not appropriate.

Original Text:

The ability of At90 RNA to repress translation was abolished by treatment with LiCl (**Figure 6c**), which correlated with destabilization of RG4 structures (**Figure 6b**).

Revised Text:

The ability of At90 RNA to repress translation was markedly reduced by treatment with LiCl (**Figure 6c**), which correlated with destabilization of RG4 structures (**Figure 6b**).

10) Since the 5'tRNAAla half is not the only tRF produced upon stress that can inhibit translation by the mechanism described herein (e.g. 5'tRNA-Cys half; ref. 21), have the authors ever thought about the possibility that in vivo different tRFs form a heterogenic intermolecular RG4 structure that might represent the functional complex?

This is a very interesting possibility. We have completed a set of experiments which suggest that this is a possibility. The results can be found in Supplemental figure 1b-c. We equilibrated biotinylated 5'Cys with truncated 5'Ala. The truncations were used so that we could distinguish between 5'Cys and 5'Ala on gels. We previously showed that the truncations used retained their activity (ref. 21). We recovered biotinylated RNA (5'Cys), eluted and assayed for the presence of 5'Ala. Indeed, 5'Cys was able to pulldown 5'Ala while non-TOG containing control RNA was not.

We added paragraph describing this experiment and its results in the text

Reviewer #2 (Remarks to the Author):

The manuscript by Lyons et al investigates the contribution of RNA G-quadruplex formed in tRNA fragments in translation inhibition.

This article follows from two publications by the same groups that i) showed that tRNAs containing an 5' oligoguanine repress translation (Mol Cell, 2011) and ii) that these tRNA 5' oligoguanine regions are able to adopt a G-quadruplex structure (PNAS, 2014). In this manuscript, the authors nicely combined various methods to further characterize the contribution of these G-quadruplexes in translation repression and stress granule formation. The manuscript is clearly presented and convincingly demonstrates that the G-quadruplexes formed in these tRNAs are crucial for translation repression and stress granule formation.

We thank this reviewer for the positive evaluation of our work and suggestions to improve it further

The authors should consider the following points:

1- Lanes 189-192: The authors are correct that the CD spectrum observed corresponds to a parallel G4. However, stem-loop RNAs display a very similar CD signature. Since the tRNA-Ala also adopts a stem-loop structure (as demonstrated by the NMR data), the CD spectra can not be unambiguously attributed to a parallel RNA G4. However, the NMR data presented in Figure 2a-c are convincing.

We were aware that the use of CD spectroscopy to determine the topology of RNA G-quadruplex can be problematic, however, we used CD experiment not to confirm 5'tiRNA^{Ala} quadruplex formation but to exclude the possibility of antiparallel topology. In order to make it more clear, the paragraph on the use of CD was modified as follows:

Original text:

CD spectra recorded for 5'tiRNA^{Ala} in the presence of K⁺, Na⁺ or Li⁺ ions displayed positive band at ~265 nm and small negative band at 240 nm (**Figure 2d**) suggesting that 5'tiRNA^{Ala} adopts the characteristic of RG4 parallel topology as for the antiparallel quadruplexes predominant positive peak at 295 nm would be expected. Although relatively little information on the 5'tiRNA^{Ala} structure is

available from CD data, nevertheless, together with the NMR data it provides supporting arguments for the presence of parallel RG4 motifs in K⁺ and Na⁺ buffers.

Revised text:

G-quadruplexes can adopt either a parallel topology, in which the backbone of each nucleic acid strand of the G-quadruplex run in the same direction, or an anti-parallel topology in which the backbones run in opposing directions. All RNA quadruplex structures currently reported exhibit a parallel topology, with one recent exception, that is, the Spinach aptamer (*Huang, H., Suslov, N.B., Li N.-S., Shelke, S. A., Evans, M. E., Koldobskaya, Y., Rice, P. A. & Piccirilli J. A., Nat. Chem. Biol. 10, 686–691 (2014)*). The use of CD spectroscopy to determine the topology of DNA G-quadruplexes is well established (*Randazzo, A., Spada, G.P. & Webba da Silva, M. Circular dichroism of quadruplex structures. Top. Curr. Chem., 330, 67–86 (2013)*), however, its use in studies of RNA G-quadruplexes can be problematic because CD spectra of RNA parallel G-quadruplexes are very similar to those of A-form structures. Nevertheless, being aware of this ambiguity, we used CD experiment mainly to exclude the possibility of antiparallel topology of RG4 quadruplex. The typical spectra of antiparallel quadruplexes have a negative band at 260 nm and positive band at 295 nm. The CD spectra recorded for 5'tiRNA^{Ala} in the presence of K⁺, Na⁺ or Li⁺ ions displayed only a positive band at ~265 nm and small negative band at 240 nm (**Figure 2d**). No positive peak at 295 nm was detected implying that RG4 5'tiRNA^{Ala} adopts a parallel structure.

2- Lanes 261-265: I was surprised to read that tiRNAs (natural and deaza) can be efficiently transfected into cells. This is very important and suggests that 7-deaza-modified RNAs can be investigated directly in cells, opening new opportunities of studying G4s in cells. I wonder how the authors confirmed that these RNAs were indeed successfully transfected into cells. The results corresponding to 5'Ala-7daG RNA presented in Figure 5b could also be interpreted by a failure to transfect the modified RNA in cells.

This is an excellent point and as such we have addressed this in the revised text. We transfected biotinylated tiRNAs and control RNAs into cells as done for SG induction. After transfection, we extracted RNAs, ran them on an acrylamide gel, transferred to nylon membrane and probed for the biotin with streptavidin-HRP. There was no difference between the WT and 7-deazaguanine modified 5'Ala. This new data can be found in Supplemental Figure 3. Moreover, it also agree with are

previously published data (Emara et al. JBC 2010, Supplementary figure 1) where we show the similar transfection efficiency of various biotinylated tiRNAs and control oligos using Immunofluorescence Microscopy.

Added to the text:

It is possible that the RG4 structure provides additional stability to 5'tiRNA^{Ala}(WT) that is lost in the 5'tiRNA^{Ala}(2x7-daG) upon transfection into cells. To account for this possibility, we extracted RNA from cells post-transfection and assayed for the presence of transfected tiRNAs. We found no differences between 5'tiRNA^{Ala}(WT) and 5'tiRNA^{Ala}(2x7-daG (**Figure S3a**).

3- Lanes 297-299, Figure 6c: I do not agree with the authors that At68B has no effect on translation. The figure suggests that At68B behaves in a very similar manner as At90.

We apologize for the confusion. We did not intend to suggest that At68B has no effect on translation. We aimed to highlight that equilibration in G4 disassembling ionic conditions did not alter their translation inhibiting activities.

Original Text:

At68B and At112A were unaffected in their ability to inhibit translation in RG4 destabilizing conditions, suggesting they utilize an alternative, non-RG4 pathway to inhibit translation

Revised text:

However, regardless of ionic conditions, At68B and At112A retained their translation inhibitory activities, suggesting they utilize an alternative, non-RG4 pathway to inhibit translation.

4- Lanes 313-320: The authors refer extensively to their previous article published in Mol Cell (2011) but less so to their previous article in PNAS (2014). The PNAS paper already demonstrated that tiRNAs can adopt G4 structures and that these structures were crucial for translation inhibition. It seems to me that this manuscript is a follow up study of their PNAS paper and therefore results from the PNAS paper should be discussed in this manuscript.

We agree that our previous publication should have been referenced more fully. We have added several references to this paper and multiple sentences highlighting it where appropriate. However, it is worth pointing out that we never completed any functional studies on the requirement of G4

structures in Ivanov et al. (2014). It is true that though “in solution NMM Staining”, we showed that 5'tiRNA^{Ala} had affinity for NMM, which strongly suggested G4-like structures were present. However, and in isolation, ligand binding assay can be false positive (discussed in our recent review (Fay et al. JMB 2017). We did not follow up experiments showing how the G4 was formed or manipulations of G4 to show any functional requirement for G4 in activity. We did show correlative data that other G4 forming nucleic acids had similar effects on cell viability.

Minor points:

1- Lane 61: The sentence refers to DNA G4s but Reference 9 refers to RNA G4s. Reference 9 should be changed to Biffi et al, Nat. Chem., 2013.

Changed

2- Lane 265: Figure 5d should not be cited here.

We did intend to refer to Figure 5d in this sentence, however, we recognize that our wording was unclear. We have revised the sentence to more clearly indicate to which data we are referring.

Original Text:

This correlated with the failure of non-G4-forming 5'tiRNA^{Ala} to interact with YB-1 (**Figure 5c**) and displace eIF4E/eIF4G complexes from m⁷GTP agarose (**Figure 5d and S3a**).

Revised Text:

This correlated with the failure of non-G4-forming 5'tiRNA^{Ala} to interact with YB-1 and displace eIF4E/eIF4G complexes from m⁷GTP agarose (**Figure 5c, Figure 5d and S3a**).

3- Lane 271: Figure 5d should be cited in place of Figure 4d

We have altered the text to refer to the proper figure.

Revised Text:

While 5'tiRNA^{Ala}(WT) readily displaced eIF4G and a portion of eIF4E, 5'tiRNA^{Ala}(2x-7daG) or control RNAs were unable to do so (**Figure 5d & S3b**).

Reviewer #3 (Remarks to the Author):

In this manuscript, Lyons et al. characterize an RNA G-quadruplex formed from tRNA fragments. The work focuses on 5'tiRNA^{Ala} which the authors purport forms a tetramolecular G4 and poses regulatory functions. This manuscript shows that tiRNA is able to form intermolecular G4 in vitro, and proposes a structure model for the inter-RNA-G4. A combination of gel-based and biophysical experiments were conducted. However, although the idea is interesting, similar results were published in their 2014 PNAS, with some data shown in the supporting information of PNAS. I am not sure whether this manuscript contains enough new information that warrants publication in Nature Comm.

We thank reviewer for his comment. The PNAS paper only suggested that tiRNAs can form G-quadruplex structures and was based on NMM assay only, which is not sufficient. The focus of that paper was DNA form of tiRNAs (so called tiDNAs). Here we used other assays and methods to demonstrate that tiRNAs assemble RNA G4s. We think that our current work unambiguously demonstrates that tiRNAs form tetramolecular G4s, which have never been shown for other RNAs before.

Using in vitro translation and in vivo reporter assay, the authors show that this inter-RNA-G4 can represses mRNA translation by displacing eIF4 complex from the mRNA 5'-capping structure. It would be interesting to understand how and why tetra-G4 can compete with the original substrate of eIF4.

This is an interesting question. We are currently started to work on addressing it. However, we feel that this mechanistically oriented work is beyond the scope of the current manuscript.

Ctrl1 and Ctrl2 in figure 5c seem quite different. The author should provide detailed sequences for these controls.

We have now added this information along with the sequences of all other oligonucleotides to the materials and methods sections.

New Text:

Oligonucleotides

All oligonucleotides were synthesized and purified by Integrated DNA Technologies.

Ctrl1: Phospho-UGAAGGGUUUUUUGUGUCUCUAUUUCCUUC-Biotin,

Ctrl2: Phospho-GCAUUCACUUGGAUAGUAAAUCCAAGCUGAA-Biotin,

5'tiRNA^{Ala}: Phospho-GGGGGUGUAGCUCAGUGGUAGAGCGCGUGC-Biotin,

5'tiRNA^{Ala} (U4G): Phospho-UGGGGUGUAGCUCAGUGGUAGAGCGCGUGC-Biotin,

5'tiRNA^{Ala} (4G): Phospho-GGGGUGUAGCUCAGUGGUAGAGCGCGUGC-Biotin,

5'tiRNA^{Ala} (UU3G): Phospho-UUGGGUGUAGCUCAGUGGUAGAGCGCGUGC-Biotin,

5'tiRNA^{Ala} (3G): Phospho-GGGUGUAGCUCAGUGGUAGAGCGCGUGC-Biotin,

5'tiRNA^{Ala} (AA3G): Phospho-AAGGGUGUAGCUCAGUGGUAGAGCGCGUGC-Biotin,

5'tiRNA^{Met}: Phospho-GCCUCGUUAGCGCAGUAGGUAGCGCGUCAGU-Biotin,

5'tiRNA^{Met} (5G): Phospho-GGGGGUUAGCGCAGUAGGUAGCGCGUCAGU-Biotin

As the work is more pertinent towards the how formation of tetramolecular G-quadruplex structure in the 5'tiRNA Ala sequence, authors should have explained more details about the structure of G-quadruplex formed. How do the authors determine which guanines are involved in the G4 formation? How is the parallel G4 structure model derived?

Thank you for this request. Following the Reviewer's suggestion we have explained more precisely the arguments that were used to propose RG4 model, and this improves readability for general

scientific audience and not only structural biologists. The paragraph was modified as follows:

Original text:

Information regarding a number of G-quartets could be deduced from ^1H NMR spectrum recorded at 55 °C in solution containing Na^+ ions (**Figure 2b**). In this spectrum only five well resolved resonances were observed indicating the presence of a quadruplex with five G-quartets. Taken together spectroscopic data provide strong evidence that in K^+ and Na^+ solutions hairpin form of $5'\text{tiRNA}^{\text{Ala}}$ exists in equilibrium with a highly symmetric, parallel tetramolecular RG4 with five G-quartets (**Figure 1b and Figure 3a**).

Revised text:

A typical G-quadruplex stem is formed from at least two stacked G-quartets. In a structure of RG4 $5'\text{tiRNA}^{\text{Ala}}$ information regarding a number of stacked G-quartets was deduced from ^1H NMR spectrum recorded at 55 °C in solution containing Na^+ ions (**Figure 2b**). In this spectrum only five well resolved resonances were observed suggesting that 5'TOG motif is engaged in quadruplex formation. Although the number of RNA strands involved in RG4 formation could not be directly inferred from NMR spectra, however, gel retardation experiments demonstrated that the slowly migrating band of $5'\text{tiRNA}^{\text{Ala}}$ was a tetramer. Therefore, the only possible interpretation of the NMR data is that RG4 $5'\text{tiRNA}^{\text{Ala}}$ adopts highly symmetric, parallel tetramolecular quadruplex with five G-quartets (**Figure 3a**).

In summary, our data indicate that $5'\text{tiRNA}^{\text{Ala}}$ can adopt two conformations, an intramolecular hairpin structure (**Figure 1b**) and tetrameric RG4 structure (**Figure 3a**). In K^+ and Na^+ solutions equilibrium between hairpin and RG4 structures exists, however, only hairpin structure forms in the presence of Li^+ ions.

How were assignments achieved in Figure S1?

To assign resonances in 2D NOESY spectrum shown in Figure S1 we followed well established procedures described in: *Varani,G., Aboul-ela,F. and Allain,F.H.-T. (1996) NMR investigation of RNA structure. Prog. Nucl. Magn. Reson. Spectrosc., 29, 51–127*. Accordingly, the legend to Figure S1 was modified as follows:

Original text:

Figure S1. Imino-imino and imino-amino region of 2D NOESY spectrum (250 ms mixing time) of 5'tiRNA^{Ala} in the presence of 150 mM NaCl, 10 mM sodium phosphate and 0.1 mM EDTA, pH 6.8 at 10 °C. G-U base-pair was inferred from strong NOE between imino G-H1 and U-H3 hydrogens.

Revised text:

Figure S1. Imino-imino and imino-amino region of 2D NOESY spectrum (250 ms mixing time) of 5'tiRNA^{Ala} in the presence of 150 mM NaCl, 10 mM sodium phosphate and 0.1 mM EDTA, pH 6.8 at 10 °C. The assignment of resonances was obtained based on the observation of NOE cross-peaks characteristic of imino protons involved in G-C and A-U Watson-Crick base pairs formation. G-U base-pair was inferred from strong NOE between imino G-H1 and U-H3 hydrogens.

5'tiRNA sequence mentioned in the Fig. 2b and Fig. 6a are different, which one is the correct one? What are the full-length sequences of the tiRNAs used in this study?

We apologize for the confusion. There was a nucleotide deleted in figure 1b. This has been corrected in the current version.

We have now added this information along with the sequences of all other oligonucleotides to the materials and methods sections.

New Text:

Oligonucleotides

All oligonucleotides were synthesized and purified by Integrated DNA Technologies.

Ctrl1: Phospho-UGAAGGGUUUUUUGUGUCUCUAUUUCCUUC-Biotin,

Ctrl2: Phospho-GCAUUCACUUGGAUAGUAAAUCCAAGCUGAA-Biotin,

5'tiRNA^{Ala}: Phospho-GGGGGUGUAGCUCAGUGGUAGAGCGCGUGC-Biotin,

5'tiRNA^{Ala} (U4G): Phospho-UGGGGUGUAGCUCAGUGGUAGAGCGCGUGC-Biotin,

5'tiRNA^{Ala} (4G): Phospho-GGGGUGUAGCUCAGUGGUAGAGCGCGUGC-Biotin,

5'tiRNA^{Ala} (UU3G): Phospho-UUGGGUGUAGCUCAGUGGUAGAGCGCGUGC-Biotin,

5'tiRNA^{Ala} (3G): Phospho-GGGUGUAGCUCAGUGGUAGAGCGCGUGC-Biotin,

5'tiRNA^{Ala} (AA3G): Phospho-AAGGGUGUAGCUCAGUGGUAGAGCGCGUGC-Biotin,

5'tiRNA^{Met}: Phospho-GCCUCGUUAGCGCAGUAGGUAGCGCGUCAGU-Biotin,

5'tiRNA^{Met} (5G): Phospho-GGGGGUUAGCGCAGUAGGUAGCGCGUCAGU-Biotin

The selected tRNA sequence has both stem-loop region and extended 5'-guanine rich sequence, how this will be shown in CD experiments was not explained. If the four tRNA molecules form tetramolecular structure, what happen to the stem and loop region?

See the response to the comment #1 of the second Referee.

The section title "Tetramolecular RG4-tiRNAs are present and functioning in plants" is misleading, as it may imply an in-cell or in-organism experiment.

Great suggestion, whave altered the heading to soften the statement.

Original text:

Tetramolecular RG4-tiRNAs are present and functioning in plants

Revised text:

Tetramolecular RG4-tiRNAs are evolutionarily conserved

REVIEWERS' COMMENTS:

Reviewer #1 (Remarks to the Author):

The authors have adequately addressed all of my minor points and I thus recommend accepting the revised version of this manuscript for publication.

Reviewer #2 (Remarks to the Author):

The authors have satisfactorily addressed all my concerns in the revised version of their manuscript.